# EHRDiff : Exploring Realistic EHR Synthesis with Diffusion Models

## Abstract

Electronic health records (EHR) contain a wealth of biomedical information, serving as valuable resources for the development of precision medicine systems. However, privacy concerns have resulted in limited access to high-quality and large-scale EHR data for researchers, impeding progress in methodological development. Recent research has delved into synthesizing realistic EHR data through generative modeling techniques, where a majority of proposed methods relied on generative adversarial networks (GAN) and their variants for EHR synthesis. Despite GAN-based methods attaining state-of-the-art performance in generating EHR data, these approaches are difficult to train and prone to mode collapse. Recently introduced in generative modeling, diffusion models have established cutting-edge performance in image generation, but their efficacy in EHR data synthesis remains largely unexplored. In this study, we investigate the potential of diffusion models for EHR data synthesis and introduce a novel method, EHRDIFF . Through extensive experiments, EHRDIFF establishes new state-of-the-art quality for synthetic EHR data, protecting private information in the meanwhile.

## 1 Introduction

Electronic health records (EHR) contain vast biomedical knowledge. EHR data may enable the development of state-of-the-art computational biomedical methods for dynamical treatment (Sonabend et al., 2020), differentiable diagnosis (Yuan & Yu, 2021), rare genetic disease identification (Alsentzer et al., 2022), etc. However, EHRs contain sensitive patients' private health information. The real-world EHRs need de-identification before publicly accessible (Johnson et al., 2016; 2023). The de-identification process uses automatic algorithms and requires tedious thorough human reviewing. Pending releasing approval can take months out of legal or ethical concerns (Hodge et al., 1999). Such circumstances limit the open-resourcing of rich EHR data, hence impeding the advancement of precision medicine methodologies. To mitigate the issue of limited publicly available EHR data, researchers alternatively explored generating synthetic EHR data (Choi et al., 2017; Walonoski et al., 2017). Realistic synthetic EHR generation has recently become a research field of medical informatics.

A line of works approached EHR data synthesis through generative modeling techniques, where they trained generative models on limited real EHR data to generate synthetic EHR data. Recent research developed variants of auto-encoders (Vincent et al., 2008; Biswal et al., 2020) or generative adversarial networks (GAN) (Goodfellow et al., 2014; Choi et al., 2017). The majority of EHR data synthesis methodologies have relied on GAN (Choi et al., 2017; Baowaly et al., 2018; Zhang et al., 2019; Yan et al., 2020). Although GAN-based methods achieved state-of-the-art performance with respect to synthetic EHR quality and privacy preservation, they suffer from training instability and mode collapse (Che et al., 2017). Previous research proposed different techniques to mitigate the problem, while as shown in our experiments, GAN-based methods still are prone to such problems, resulting in unsatisfactory synthetic data quality. This may raise concerns when developing real-world systems using synthetic EHR data from GAN-based methods.

Most recently, novel diffusion models (Sohl-Dickstein et al., 2015) in generative modeling have been proposed and have achieved cutting-edge generation performance in the field of vision (Ho et al., 2020; Song et al.,

2021), audio (Kong et al., 2021), or texts (Li et al., 2022; Gong et al., 2023; Yuan et al., 2022). Many variants of diffusion models have surpassed the generation performance of GANs in sample quality and diversity. In general, starting from random noise features, diffusion models use a trained denoising distribution to gradually remove noise from the features and ultimately generate realistic synthetic features. The efficacy of diffusion models on realistic EHR synthesis is rarely studied. Considering the superior performance of diffusion models in other domains, our work explores the synthesizing performance of such techniques on EHR data. We design and introduce EHRDIFF , a diffusion-based EHR synthesizing model.

Our work conducts comprehensive experiments using publicly available real EHR data and compares the effectiveness of EHRDIFF against several other GAN-based EHR data synthesizing methods. We provide empirical evidence that EHRDIFF is capable of generating synthetic EHR data with a high degree of quality. Additionally, our findings reveal that the synthetic EHR data produced by EHRDIFF is of superior quality compared to those generated by GAN-based models, and it is more consistent with the distribution of real-world EHR data.

Our research has two primary contributions: firstly, we introduce the use of diffusion models to the realm of realistic EHR synthesis and propose a diffusion-based method called EHRDIFF . Secondly, through extensive experimentation on publicly available EHR data, we demonstrate the superior quality of synthetic EHR data generated by EHRDIFF in comparison to GAN-based EHR synthesizing methods. Furthermore, the synthetic EHR data generated by EHRDIFF exhibits excellent correlation with real-world EHR data.

Our work is summarized as two following contributions:

1. We introduce diffusion models to the fields of EHR data synthesis and propose a diffusion-based method called EHRDIFF .

2. Through extensive experiments on publicly available real EHR data, we empirically demonstrate the superior generation quality of EHRDIFF over GAN-based EHR synthesis methods. EHRDIFF can safeguard private information in real EHR for training.

3. Compared to existing GAN-based methods, EHRDIFF also shows superior efficacy for synthesizing various EHR feature formats, including categorical, continuous, and time-series features.

## 2 Related Works

### 2.1 EHR data synthesis

In the literature on EHR synthesis, researchers are usually concerned with the generation of discrete code features such as ICD codes rather than clinical narratives. Researchers have developed various methods to generate synthetic EHR data. Early works were usually disease-specific or covered a limited number of diseases. Buczak et al. (2010) developed a method that generates EHR related to tularemia and the features in synthetic EHR data are generated based on similar real-world EHR which is inflexible and prone to privacy leakage. Walonoski et al. (2017) developed a software named Synthea which generates synthetic EHR based on publicly available data. They build generation workflows based on biomedical knowledge and real-world feature statistics. Synthea only covered the 20 most common conditions.

Recently, researchers mainly applied generative modeling methods for EHR synthesis (Ghosheh et al., 2022). Medical GAN (medGAN) (Choi et al., 2017) introduced GAN to EHR synthesis. medGAN can generate synthetic EHR data with good quality and is free of tedious feature engineering. Following medGAN, various GAN-based methods are proposed, such as medBGAN (Baowaly et al., 2018), EHRWGAN (Zhang et al., 2019), CorGAN (Torfi & Fox, 2020a), etc. These GAN-based methods advance synthetic EHR to higher quality. However, a common drawback of GAN-based methods is that these methods suffer from the mode collapse phenomenon which results in a circumstance where a GAN-based model is only capable to generate only a few modes of real data distribution (Thanh-Tung et al., 2018). To mitigate the problem, GAN methods for EHR generation rely on pre-trained auto-encoders to reduce the feature dimensions for training stability. However, inappropriate hyper-parameter choices and autoencoder pre-training will lead to sharp degradation of synthetic EHR quality or even failure to generate realistic data. There is also research that uses GAN-based

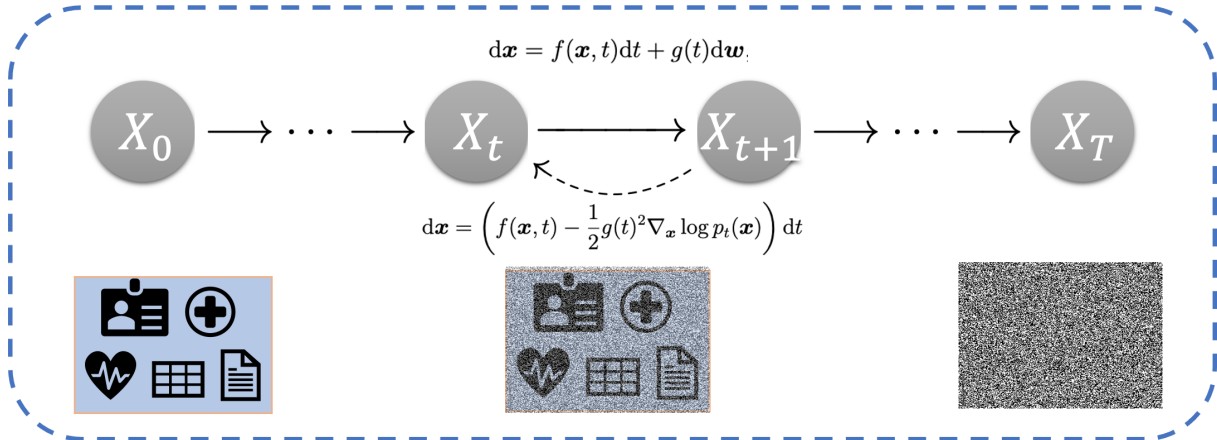

$$d\boldsymbol{x} = f(\boldsymbol{x}, t)dt + g(t)d\boldsymbol{w}$$

$$d\boldsymbol{x} = \left( f(\boldsymbol{x}, t) - \frac{1}{2}g(t)^2 \nabla_{\boldsymbol{x}} \log p_t(\boldsymbol{x}) \right) dt$$

Figure 1: An overview of proposed EHRDIFF .

models for conditional synthetic EHR generation to model the temporal structure of real EHR data (Zhang et al., 2020a). Since diffusion models are less studied in EHR synthesis, we focus on the unconditional generation of EHR and leave modeling conditional temporal structure with diffusion models to future works.

Besides GAN-based methods, there also exists research that explores generating synthetic EHR data through variational auto-encoders (Biswal et al., 2020) or language models (Wang & Sun, 2022). Concurrently, MedDiff (He et al., 2023) is proposed and explores diffusion models for synthetic EHR generation, and they propose a new sampling technique without which the diffusion model fails to generate high-quality EHRs. In our work, we explore the direct implementation of diffusion models to generate synthetic EHR data.

## 2.2 Diffusion Models

Diffusion models are formulated with forward and reverse processes. The forward process corrupts real-world data by gradually injecting noise, and harvesting training data with different noise levels for a denoising distribution, while the reverse process generates realistic data by removing noise using the denoising distribution. Diffusion models are first proposed and theoretically supported (Sohl-Dickstein et al., 2015). DDPM (Ho et al., 2020) and NCSN (Song et al., 2021) discover the superior capability in image generation, and diffusion models become a focused research direction since then. Recent research generalizes diffusion models to the synthesis of other data modalities and achieves excellent performance (Li et al., 2022; Kong et al., 2021). Our work for the first time introduces diffusion models to realistic EHR synthesis.

## 3 Method

In this section, we give an introduction to the problem formulation of realistic EHR synthesis and the technical details of our proposed EHRDIFF . An overview of EHRDIFF is depicted in Figure 1.

## 3.1 Problem Formulation

Following previous research (Choi et al., 2017; Baowaly et al., 2018), we focus on the synthesis of code features in EHR data. We assume a set of codes $\mathcal{C}$ of interest, and one EHR sample is encoded as a fixed-size feature vector $\boldsymbol{x}_0 \in \{0, 1\}^{|\mathcal{C}|}$. The $i$-th dimension represents the occurrence of the corresponding code feature, where 1 stands for occurrence and 0 otherwise. In EHRDIFF , we treat the binaries as real numbers and directly apply forward and backward diffusion processed upon $\boldsymbol{x}_0$.

### 3.2 EHRDiff

Generally, diffusion models are characterized by forward and reverse Markov processes with latent variables. As demonstrated by Song et al. (2021), the forward and reverse processes can be described by stochastic differential equations (SDE). In EHRDIFF , we use differential equations to describe the processes. The general SDE form for modeling the forward process follows:

$$\mathrm{d}\boldsymbol{x} = f(\boldsymbol{x},t)\mathrm{d}t + g(t)\mathrm{d}\boldsymbol{w}, \tag{1}$$

where $\boldsymbol{x}$ represents data points, $\boldsymbol{w}$ represents standard Wiener process, $t$ is diffusion time and ranges from 0 to $T$. At $t = 0$, $\boldsymbol{x}$ follows real data distribution while at $t = T$, $\boldsymbol{x}$ asymptotically follows a random Gaussian distribution. Functions $f$ and $g$ define the sample corruption pattern and the level of injected noises, respectively. $f$ and $g$ together corrupt real-world data to random noise, following the SDE above. Based on the forward SDE, we can derive the SDE for the reverse process as:

$$\mathrm{d}\boldsymbol{x} = \left( f(\boldsymbol{x},t) - g^2(t)\nabla_{\boldsymbol{x}}\log p_t(\boldsymbol{x}) \right)\mathrm{d}t + g(t)\mathrm{d}\boldsymbol{w}, \tag{2}$$

where $p_t(\boldsymbol{x})$ is the marginal density of $\boldsymbol{x}$ at time $t$. Therefore, to generate data from random noise, we need to learn the score function $\nabla_{\boldsymbol{x}}\log p_t(\boldsymbol{x})$ with score matching. The score function indicates a vector field in which the direction is pointed to the high-density data area. The objective is to estimate the score function and is formulated as:

$$\min_{\theta} \mathbb{E}_{p(\boldsymbol{x}_0)p_{\sigma_t}(\boldsymbol{x}|\boldsymbol{x}_0)} \left[ \|s_\theta(\boldsymbol{x}) - \nabla_{\boldsymbol{x}}\log p_{\sigma_t}(\boldsymbol{x}|\boldsymbol{x}_0)\|_2^2 \right], \tag{3}$$

where $p(\boldsymbol{x_0})$ is the density for real data, $s_\theta$ is the score function parameterized by $\theta$, $\nabla_{\boldsymbol{x}}$ is the operator of derivation with respect to $\boldsymbol{x}$, $p_{\sigma_t}(\boldsymbol{x}|\boldsymbol{x}_0)$ is named the perturbation kernel corresponding to the SDE. The kernel represents the conditional density of noisy sample $\boldsymbol{x}$ at noise level $\sigma_t$ and is formulated as:

$$p_{\sigma_t}(\boldsymbol{x}|\boldsymbol{x}_0) = \mathcal{N}(h(t)\boldsymbol{x}_0, h(t)^2\sigma_t^2\boldsymbol{I}), \tag{4}$$

where $h(t)$ and $\sigma_t$ are reformulated from $f$ and $g$ for concise notations:

$$h(t) = \exp\left( \int_0^t f(\xi)\mathrm{d}\xi \right),$$

$$\sigma_t = \sigma(t) = \sqrt{\int_0^t \frac{g(\xi)^2}{h(\xi)^2}\mathrm{d}\xi}.$$

Note that we change the subscript of $p_t$ to $p_{\sigma_t}$ in Equation 3, since $t$ controls the noisy sample distribution through the noise level $\sigma_t$. In EHRDIFF , we follow the perturbation kernel design from previous research (Karras et al., 2022) in which $h(t) = 1$ and $\sigma_t = t$. Therefore $\nabla_{\boldsymbol{x}}\log p_{\sigma_t}(\boldsymbol{x}|\boldsymbol{x_0}) = -\frac{\boldsymbol{x}-\boldsymbol{x}_0}{\sigma_t^2}$ and we reparameterize $s_\theta(\boldsymbol{x}) = -\frac{\boldsymbol{x}-D_\theta(\boldsymbol{x},\sigma_t)}{\sigma_t^2}$, then the objective can be derived as:

$$\min_{\theta} \mathbb{E}_{p(\boldsymbol{x_0})p_{\sigma_t}(\boldsymbol{x}|\boldsymbol{x_0})} \left[ \left\| -\frac{x - D_\theta(x,\sigma_t)}{\sigma_t^2} + \frac{x - x_0}{\sigma_t^2} \right\|_2^2 \right], \tag{5}$$

$$\min_{\theta} \mathbb{E}_{p(\boldsymbol{x_0})p_{\sigma_t}(\boldsymbol{x}|\boldsymbol{x_0})} \left[ \left\| \frac{D_\theta(x,\sigma_t) - x_0}{\sigma_t^2} \right\|_2^2 \right], \tag{6}$$

and with further simplification, the final objective becomes:

$$\min_{\theta} \mathbb{E}_{p(\boldsymbol{x_0})p_{\sigma_t}(\boldsymbol{x}|\boldsymbol{x_0})} \left[ \|D_\theta(x,\sigma_t) - x_0\|_2^2 \right], \tag{7}$$

where $D(x,\sigma_t)$ is the denoising function that predicts the denoised samples based on noisy ones. With such parameterization, the score function $\nabla_{\boldsymbol{x}}\log p_{\sigma_t}(\boldsymbol{x})$ can be recovered by:

$$\nabla_{\boldsymbol{x}}\log p_{\sigma_t}(\boldsymbol{x}) = -\frac{x - D_\theta(x,\sigma_t)}{\sigma_t^2}. \tag{8}$$

---
**Algorithm 1** Heun's 2nd Method for Sampling

---
**Input:** Time Step $t_i$ and noise level $\sigma_{t_i}$
1: Using Equation 8 and 10, calculate the derivative $\boldsymbol{g}_{t_i} = \mathrm{d}\boldsymbol{x}/\mathrm{d}t$:
   $\boldsymbol{g}_{t_i} = t_i^{-1}\boldsymbol{x}_{t_i} - t_i^{-1}D(\boldsymbol{x}_{t_i}; \sigma_{t_i})$,
2: Get intermediate $\tilde{\boldsymbol{x}}_{t_{i+1}}$ by taking Euler step:
   $\tilde{\boldsymbol{x}}_{t_{i+1}} = \boldsymbol{g}_{t_i}(t_{i+1} - t_i) + \boldsymbol{x}_{t_i}$,
3: Calculate the gradient correction $\tilde{\boldsymbol{g}}_{t_i}$:
   $\tilde{\boldsymbol{g}}_{t_i} = t_{i+1}^{-1}\tilde{\boldsymbol{x}}_{t_{i+1}} - t_{i+1}^{-1}D(\tilde{\boldsymbol{x}}_{t_{i+1}}; \sigma_{t_{i+1}})$,
4: Get next time step sample $\boldsymbol{x}_{t_{i+1}}$:
   $\boldsymbol{x}_{i+1} = \boldsymbol{x}_i + (t_{i+1} - t_i)(\frac{1}{2}\boldsymbol{g}_i + \frac{1}{2}\tilde{\boldsymbol{g}}_{i+1})$
5: **return** $\boldsymbol{x}_{t_{i+1}}$

---

### 3.3 Deterministic Reverse Process

Diffusion models generate synthetic samples following the reverse process. Generally, the reverse processes are described by SDEs, and noise is injected in each step when solving the reverse SDE numerically. We can equivalently describe the reverse generation process with ordinary differential equations (ODE) instead of SDEs (Song et al., 2021):

$$\mathrm{d}\boldsymbol{x} = \left( f(\boldsymbol{x}, t) - \frac{1}{2}g(t)^2 \nabla_{\boldsymbol{x}} \log p_t(\boldsymbol{x}) \right) \mathrm{d}t. \tag{9}$$

The corresponding ODE is called the probability flow ODE which indicates a deterministic generation process (i.e., a deterministic numerical solution trajectory). With the aforementioned formalization of $h(t)$ and $\sigma_t$, the probability flow ODE can be rewritten as:

$$\mathrm{d}\boldsymbol{x} = -t\nabla_x \log p_{\sigma_t}(\boldsymbol{x})\mathrm{d}t. \tag{10}$$

With the learned denoising function $D_\theta(x, \sigma_t)$ and Equation 8, we can solve the ODE in Equation 10 and generate realistic synthetic EHRs from random noise.

Solving the ODE numerically requires discretization of the time step $t$ and a proper design of noise level $\sigma_t$ along the solution trajectory. As suggested in previous works, using a fixed discretization of $t$ may result in sub-optimal performance and the noise level should decrease during generation. Therefore, following previous research (Karras et al., 2022), we set the maximum and minimum of noise level as $\sigma_{\max}$ and $\sigma_{\min}$ respectively, and according to our design of $\sigma_t$, we use the following form of discretization:

$$t_i = \sigma_{t_i} = \left( (\sigma_{\max})^{\frac{1}{\rho}} + \frac{i}{N-1}\left( (\sigma_{\min})^{\frac{1}{\rho}} - (\sigma_{\max})^{\frac{1}{\rho}} \right) \right)^\rho, \tag{11}$$

where $i$'s are integers and range from 0 to $N$, $\sigma_{t_N} = 0, \sigma_{t_{N-1}} = \sigma_{\min}$, and $\rho$ controls the schedules of discretized time step $t_i$ and trades off the discretized strides $t_i - t_{i-1}$ the larger value of which indicates a larger stride near $t_0$. Numerically solving the probability flow ODE with the above-mentioned discretized time schedule leads to an approximate solution and the precision is hampered by truncation errors. In order to solve the ODE more precisely and generate synthetic EHR with higher quality, we use Heun's 2nd order method, which adds a correction updating step for each $t_i$ and alleviates the truncation errors compared to the 1st order Euler method.

We brief our 2nd order sampling method in Algorithm 1 for each time step $t_i$.

### 3.4 Design of Denoising Function

In this section, we further discuss the parameterization of the denoising function $D_\theta(x, \sigma_t)$. The denoising function $D_\theta(x, \sigma_t)$ takes the noisy sample $x$ and current noisy level $\sigma_t$ as inputs to cancel out the noise in $x$ at time step $t$. It is natural to direct model $D(x, \sigma_t)$ with neural networks, while such direct modeling may set obstacles for the training of neural networks, because, at different time steps $t$, the variance of $x$ and

scale of $\sigma_t$ are diverse. Another common practice of diffusion models is to decouple noise from input noisy sample $x$ hence $D_\theta(\boldsymbol{x}; \sigma) = \boldsymbol{x} - \sigma F_\theta(\boldsymbol{x}; \sigma)$ (Ho et al., 2020). $F_\theta(\boldsymbol{x}; \sigma)$ is modeled with neural networks and predicts the noise in $x$. Under this design, the prediction error of $F_\theta(\boldsymbol{x}; \sigma)$ may be amplified by the noise scale $\sigma$, especially when $\sigma$ is large.

Emphasizing the above problems, we use a recently proposed adaptive parameterization of $D_\theta(\boldsymbol{x}; \sigma)$ (Karras et al., 2022), where

$$D_\theta(\mathbf{x}; \sigma) = c_{\text{skip}}(\sigma)\mathbf{x} + c_{\text{out}}(\sigma)F_\theta(c_{\text{in}}(\sigma)\mathbf{x}; c_{\text{noise}}(\sigma)). \tag{12}$$

Specifically, $c_{\text{in}}(\sigma) = 1/\sqrt{\sigma^2 + \sigma_{\text{data}}^2}$ and it regulate the input to be unit variance across different noise levels. $c_{\text{out}}$ and $c_{\text{skip}}$ together set the neural model prediction to be unit variance with minimized scale $c_{\text{out}}$. Therefore, $c_{\text{out}} = \sigma\sigma_{\text{data}}/\sqrt{\sigma^2 + \sigma_{\text{data}}^2}$ and $c_{\text{skip}}(\sigma) = \sigma_{\text{data}}^2/(\sigma^2 + \sigma_{\text{data}}^2)$. $c_{\text{noise}}(\sigma) = 0.25\ln\sigma$ which is designed empirically and the principal is to constrain the input noise scale from varying immensely.

By substituting $D_\theta(\mathbf{x}; \sigma)$ in the objective Equation 7 with Equation 12, the loss function for neural model training is:

$$\mathbb{E}_{\sigma, \boldsymbol{x}, \boldsymbol{x_0}}[c_{\text{out}}(\sigma)^2 || F_\theta(c_{\text{in}}(\sigma) \cdot (\boldsymbol{x}); c_{\text{noise}}(\sigma)) - \frac{1}{c_{\text{out}}(\sigma)}(\boldsymbol{x_0} - c_{\text{skip}}(\sigma) \cdot (\boldsymbol{x}))||_2^2]. \tag{13}$$

To balance the loss at different noise levels $\sigma$, the weight $c_{\text{out}}^2(\sigma)$ is omitted in the final loss function. During training, the noise distribution is assumed to be $\ln(\sigma) \sim \mathcal{N}(P_{\text{mean}}, P_{\text{std}}^2)$, where $P_{\text{mean}}$ and $P_{\text{std}}$ are hyperparameters to be set.

## 4 Experiments

To demonstrate the effectiveness of our proposed EHRDIFF , we conduct extensive experiments evaluating the quality of synthetic EHRs and the privacy concerns of the method. We also compare EHRDIFF the several GAN-based realistic EHR synthesis methods to illustrate the performance of EHRDIFF .

### 4.1 Dataset

Many previous works use in-house EHR data which is not publicly available for method evaluation (Zhang et al., 2019; Yan et al., 2020). Such experiment designs set obstacles for later research to reproduce experiments. In this work, we use a publicly available EHR database, MIMIC-III, to evaluate EHRDIFF .

Deidentified and comprehensive clinical EHR data is integrated into MIMIC-III (Johnson et al., 2016). The patients are admitted to the intensive care units of the Beth Israel Deaconess Medical Center in Boston. For each patient's EHR, we extract the diagnosis and procedure ICD-9 code and truncate the ICD-9 code to the first three digits. This preprocessing can reduce the long-tailed distribution of the ICD-9 code distribution and results in a 1,782 code set. Therefore, the EHR for each patient is formulated as a binary vector of 1,782 dimensions. The final extracted number of EHRs is 46,520 and we randomly select 41,868 for model training while the rest are held out for evaluation.

### 4.2 Baselines

To better demonstrate EHR synthesis performance, we compare EHRDIFF to several strong baseline models as follows.

**medGAN** (Choi et al., 2017) is the first work that introduces GAN to generating realistic synthetic EHR data. Considering the obstacle of directly using GAN on generating high-dimensional binary EHR vectors, medGAN alters to a low-dimensional dense space for generation by taking advantage of pre-trained auto-encoders. The model generates a dense EHR vector and then recovers a synthetic EHR with decoders.

**medBGAN and medWGAN**   (Baowaly et al., 2018) are two improved GAN models for realistic EHR synthesis. medGAN is based on the conventional GAN model for EHR synthesis, and such a model is prone to mode collapse where GAN models may fail to learn the distribution of real-world data. medBGAN and medWGAN integrate Boundary-seeking GAN (BGAN) (Hjelm et al., 2018) and Wasserstein GAN (WGAN) (Adler & Lunz, 2018) respectively to improve the performance of medGAN and stabilize model training.

**CorGAN**   (Torfi & Fox, 2020b) is a novel work that utilizes convolutional neural networks (CNN) instead of multilayer perceptrons (MLP) to model EHR data. Specifically, they use CNN to model the autoencoder and the generative network. They empirically elucidate through experiments that CNN can perform better than the MLP in this task.

**EMR-WGAN**   (Zhang et al., 2020b) is proposed to further refine the GAN models from several perspectives. To avoid model collapse, the authors take advantage of WGAN. The most prominent feature of EMR-WGAN is that it is directly trained on the discrete EHR data, while the previous works universally use an autoencoder to first transform the raw EHR data into low-dimensional dense space. They utilize BatchNorm (Ioffe & Szegedy, 2015) for the generator and LayerNorm (Ba et al., 2016) for the discriminator to improve performance. As is shown in their experiments, these modifications significantly improve the performance of GAN.

### 4.3   Evaluation Metric

In our experiments, we evaluate the generative models' performance from two perspectives: utility and privacy (Yan et al., 2022). Utility metrics evaluate the quality of synthetic EHRs and privacy metrics assess the risk of privacy breaches. In the following metrics, we generate and use the same number of synthetic EHR samples as the number of real training EHR samples.

#### 4.3.1   Utility Metrics

We follow previous works for a set of utility metrics. The following metrics evaluate synthetic EHR quality from diverse perspectives.

**Dimension-wise distribution**   describes the feature-level resemblance between the synthetic data and the real data. The metric is widely used in previous works to investigate whether the generative model is capable to learn the high-dimensional distribution of real EHR data. For each code dimension, we calculate the empirical mean estimation for synthetic and real EHR data respectively. The mean estimation indicates the prevalence of the code. We visualize the dimension-wise distribution using scatter plots where both axes represent the prevalence of synthetic and real EHR respectively. Many codes have very low prevalence in real EHR data. The generation model may prone to mode collapse and fails to generate the codes with low prevalence. Therefore, we count the number of codes that exist in the synthetic EHR samples and dub the quantity non-zero code columns.

**Dimension-wise correlation**   measures the difference between the feature correlation matrices of real and synthetic EHR data. The $i, j$ entry of correlation matrices calculates the Pearson correlation between the $i$th and $j$th features. For both the synthetic and real EHR data, we calculate first the correlation matrices, and then the averaged absolute differences between the correlation matrices. We name this metric the correlation matrix distance (CMD).

**Dimension-wise prediction**   evaluates whether generative models capture the inherent code feature relation by designing classification tasks. Specifically, we select one of the code features to be the classification target and use the rest of the features as predictors. To harvest a balanced target distribution, we sort the code features according to the entropy $H(p)$ of code prevalence $p$, where $H(p) = -p \log(p) - (1-p) \log(1-p)$. We select the top 30 code features according to entropies and form 30 individual classification tasks. For each task, we fit a classification model with logistic regression using real training and synthetic EHR data and assess the F1 score on the preset evaluation real EHR data.

Table 1: NZC represents Non-Zero code Columns, CMD represents Correlation Matrix Distance. ↓ and ↑ indicate the respectively lower and higher numbers for better results.

|          | NZC (↑) | CMD (↓) |
|----------|---------|---------|
| medGAN   | 560     | 29.302  |
| medBGAN  | 848     | 54.833  |
| medWGAN  | 420     | 8.395   |
| CorGAN   | 799     | 11.439  |
| EMR-WGAN | 1092    | 8.089   |
| EHRDIFF  | **1742** | **7.790** |

### 4.3.2   Privacy Metrics

Generative modeling methods need real EHR data for training which raises privacy concerns among practitioners. Attackers may infer sensitive private information from trained models. Besides the utility of synthetic EHR data, we also evaluate existing models from a privacy protection perspective (Choi et al., 2017; Zhang et al., 2019; Yan et al., 2022).

**Attribute inference risk**   describes the risk that sensitive private information of real EHR training data may be exposed based on the synthetic EHR data It assumes a situation where the attackers already have several real EHR training samples with partially known features and try to infer the rest features through generated synthetic data. Specifically, we assume that attackers first use the k-nearest neighbors method to find the top $k$ most similar synthetic EHRs to each real EHRs based on the known code features, and then recover the rest of unknown code features by majority voting of $k$ similar synthetic EHRs. We set $k$ to 1 and use the most frequent 256 codes as the features known by the attackers. The metric is quantified by the prediction F1-score of the unknown code features.

**Membership inference risk**   evaluates the risk that given a set of real EHR samples, attackers may infer the samples used for training based on synthetic EHR data. We mix a subset of training real EHR data and held-out testing real EHR data to form an EHR set. For each EHR in this set, we calculate the minimum L2 distance with respect to the synthetic EHR data. The EHR whose distance is smaller than a preset threshold is predicted as the training EHR. We report the prediction F1 score to demonstrate the performance of each model under membership inference risk.

### 4.4   Implementation Detail

In our experiments on MIMIC-III, for the diffusion noise schedule, we set $\sigma_{\min}$ and $\sigma_{\max}$ to be 0.02 and 80. $\rho$ is set to 7 and the time step is discretized to $N = 32$. $P_{mean}$ is set to $-1.2$ and $P_{std}$ is set to 1.2 for noise distribution in the training process. For $F_\theta$ in Equation 12, it is parameterized by an MLP with ReLU (Agarap, 2018) activations and the hidden states are set to $[1024, 384, 384, 384, 1024]$. For the baseline methods, we follow the settings reported in their papers.

### 4.5   Results

### 4.5.1   Utility Results

Figure 2 depicts the dimension-wise prevalence distribution of synthetic EHR data against real data. The scatters from EHRDIFF are distributed more closely to the diagonal dashed line compared to other baseline models, and EHRDIFF and EMR-WGAN achieve near-perfect correlation. As shown in Table 1, EHRDIFF outperforms all baseline methods in non-zero code column number (NZC) by large margins. This shows that GAN-based baselines all suffer from model collapse to different extents. The GAN-based method of best performance, EMR-WGAN, still fails to generate 690 code features with the same number of synthetic EHR samples as the real data. Although EMR-WGAN achieves a near-perfect correlation between real and

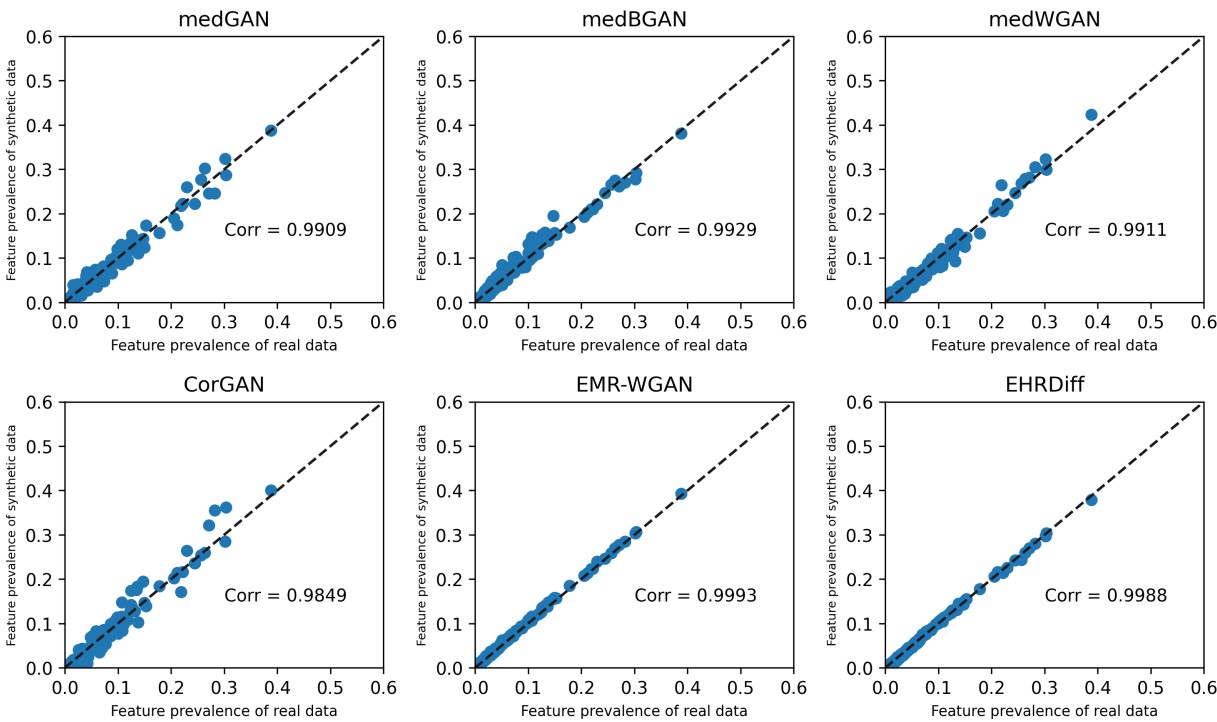

Figure 2: The dimension-wise probability scatter plot of synthetic EHR data from different generative models against real EHR data. The diagonal lines represent the perfect match of code prevalence between synthetic and real EHR data.

Table 2: The privacy assessment for each model. ↓ indicates the lower numbers for better results.

|  | Attribute Inference Risk (↓) | Membership Inference Risk (↓) |
| --- | --- | --- |
| medGAN | **0.0010** | 0.3037 |
| medBGAN | 0.0057 | 0.3036 |
| medWGAN | 0.0072 | 0.2967 |
| CorGAN | 0.0023 | **0.2393** |
| EMR-WGAN | 0.0257 | 0.2941 |
| EHRDIFF | 0.0177 | 0.2964 |

synthetic code prevalence and is slightly better than EHRDIFF , NZC demonstrates that the correlation can be biased by high prevalence features and overshadow the evaluation of low prevalence features. The results above demonstrate that EHRDIFF can better capture the code feature prevalence of the real data than the GAN-based baselines, and is free from mode collapse. The synthetic EHR data by EHRDIFF has better diversity than that by GAN-based methods.

From CMD results in Table 1, EHRDIFF surpasses all baseline models. As shown in Figure 3, the F1 score scatters of EHRDIFF are closer to the diagonal lines and achieve the highest correlation value. This means that training on synthetic EHR data by EHRDIFF can lead to more similar performance to training on real data. CMD results demonstrate that EHRDIFF can better capture the inherent pair-wise relations between code features than GAN-based methods, and Figure 3 also illustrates EHRDIFF can better model complex interactions between code features than baselines. It is also indicated that synthetic EHR data by EHRDIFF may have superior utility for developing downstream biomedical methodologies. We present more results on other utility metrics in A.

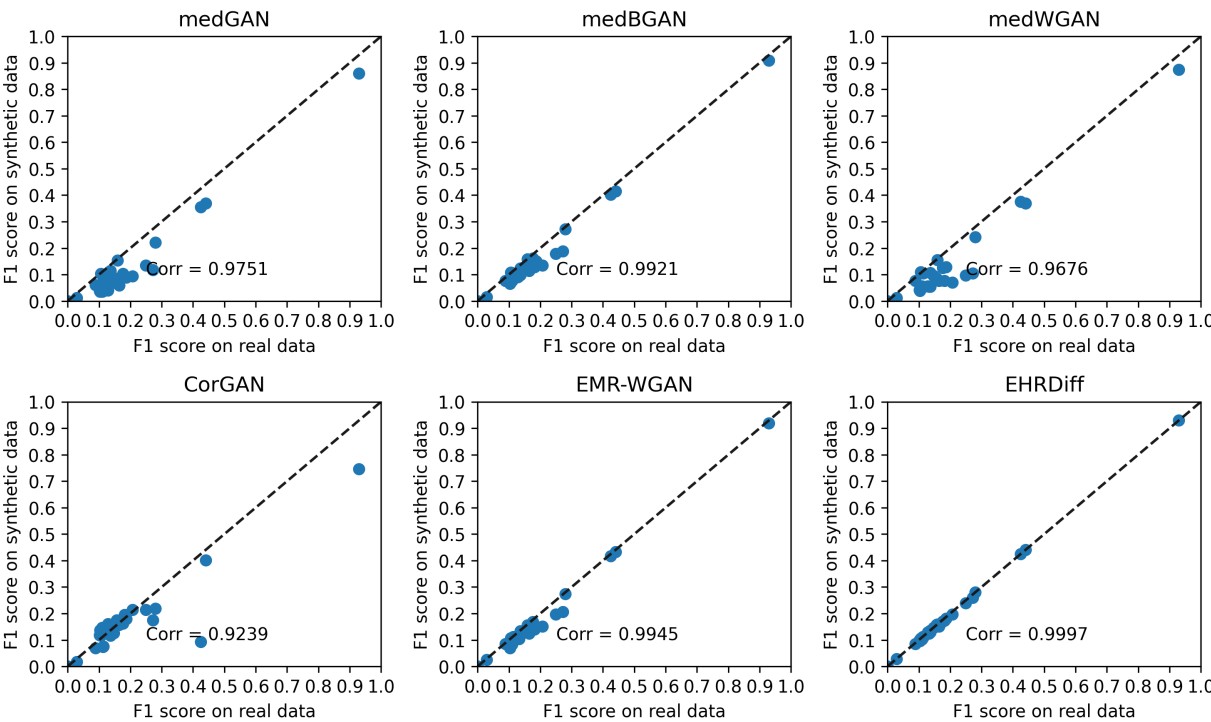

Figure 3: The dimension-wise prediction scatter plot of synthetic EHR data from different generative models against real EHR data. The diagonal lines represent the perfect match of code prediction between synthetic and real EHR data. Each scatter represents a task.

### 4.5.2   Privacy Results

In Table 2, we list the results against privacy attacks. In terms of attribute inference risk and membership inference risk, EHRDIFF achieves middling results, while medGAN and CorGAN respectively achieve the best results on attribute inference risk and membership inference risk. However, as shown in utility results, the quality of synthetic EHR data by both models is far worse than EHRDIFF . In an extreme circumstance where a generative model fails to fit the real EHR data distribution, the model may achieve perfect results on both privacy metrics, since attackers can not infer private information through synthetic data of bad quality. Therefore, there exists an implicit trade-off between utility and privacy. We suspect that medGAN and CorGAN can better safeguard privacy due to mediocre synthesis quality. When compared to EMR-WGAN which achieves the best synthesis quality among baselines, EHRDIFF surpasses EMR-WGAN on attribute inference risk and achieves on-par results in terms of membership inference risk. To conclude, EHRDIFF can well protect the sensitive private information of real EHR training data.

## 5   Discussion

### 5.1   Beyond Binary Code Features

Although most of the existing works focused on synthesizing discrete code features, real-world EHR data contains various data formats such as continuous test results values or time series of electrocardiograms (ECG). In this section, we explore extending EHRDIFF to the synthesis of EHR data other than binary codes. We use the following two datasets: CinC2012 Data and PTB-ECG Data. CinC2012 Data is a dataset for predicting the mortality of ICU patients and contains various feature formate such as categorical age, or continuous serum glucose values. PTB-ECG Data contains ECG signal data for heart disease diagnosis. Detailed introductions of both datasets are left in B. All the categorical features are converted into binary

Table 3: The AUC values on Cinc2012 Data and PTB-ECG Data.

|  | Cinc2012 Data | PTB-ECG Data |
| --- | --- | --- |
| Real | 0.8479 | 0.9963 |
| medGAN | 0.6427 | 0.7488 |
| medBGAN | 0.6550 | 0.7496 |
| medWGAN | 0.6991 | 0.7923 |
| CorGAN | 0.7659 | 0.5126 |
| EMR-WGAN | 0.7963 | 0.7843 |
| EHRDiff | **0.8375** | **0.9814** |

columns by one-hot encoding, and continuous features are normalized to values in the range of $[0.0, 1.0]$. We use set A and B in CinC2023 Data as training and held-out testing sets respectively. The PTB-ECG Data is split with a ratio of 8:2 for training and held-out testing.

## 5.2 Results

Since both datasets are designed for classification, we inspect the utilities of synthesized data by evaluating Area Under receiver operating characteristic Curve (AUC) of classifiers trained with synthetic data. We use LightGBM (Ke et al., 2017) as classifiers and train on synthetic data of the same size as real training data.

The results shown in Table 3 that classifiers trained by synthetic data from EHRDIFF achieve the highest AUC values and are consistently better than GAN-based methods, reaching 0.8375 and 0.9814 on CinC2012 Data and PTB-ECG Data respectively. They also have on-par performance with classifiers trained by real data. The results show the great utility of EHRDIFF generated EHR data, and the efficacy is consistently good across different EHR data feature formats. This demonstrates EHRDIFF is practical in real-world scenarios and can approach EHR synthesis of diverse formats. The development of downstream biomedical methodologies can benefit from synthetic EHR data by EHRDIFF , overcoming the obstacles of limited publicly available real EHR data.

## 6 Conclusion

In this work, we explore EHR data synthesis with diffusion models. We proposed EHRDIFF , a diffusion-based model, for EHR data synthesis. Through comprehensive experiments on binary code EHR data, we empirically demonstrate the superior performance in generating high-quality synthetic EHR data from multiple evaluation perspectives, setting new state-of-the-art EHR synthesis methods. In the meanwhile, we also show EHRDIFF can safeguard sensitive private information in real EHR training data. Furthermore, beyond binary code features in EHR data, the efficacy of EHRDIFF consistently excels in continuous and time-series features. EHRDIFF can help downstream biomedical methodology research overcome the obstacles of limited publicly available real EHR data.

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

Table 4: MCAD represents Medical Concept Abundance Distance. ↓ and ↑ indicate the respectively lower and higher numbers for better results.

|          | Latent Distance (↓) | MCAD (↓) |
|----------|---------------------|----------|
| medGAN   | -4.307              | 0.250    |
| medBGAN  | -4.320              | 0.112    |
| medWGAN  | **-14.771**         | 0.071    |
| CorGAN   | -7.660              | 0.145    |
| EMR-WGAN | -13.727             | 0.104    |
| EHRDIFF  | -13.849             | **0.069**|

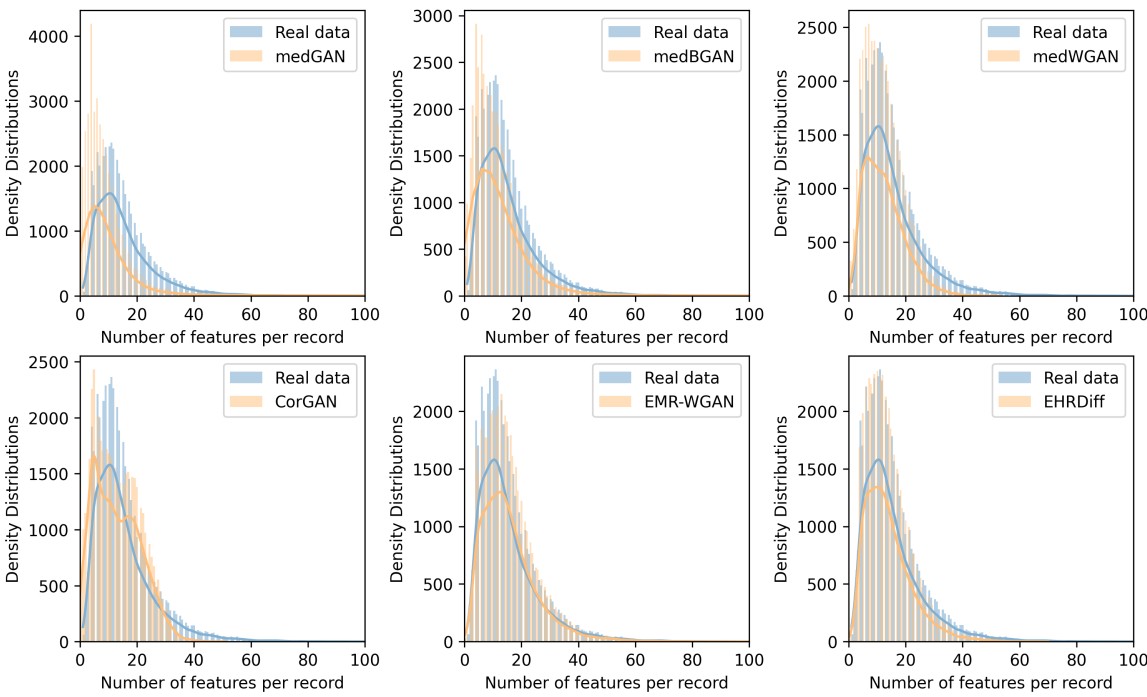

Figure 4: The histograms plot the empirical distributions of the unique code counts on the sample level. The solid lines are the kernel density estimations of the distribution.

# A  Additional Results

The quality of synthetic EHR data can be evaluated from a multifaceted perspective (Yan et al., 2022). We use additional metrics to further evaluate the synthetic EHR data on MIMIC-III.

## A.1  Latent cluster analysis

The metric evaluates the distributional difference between the synthetic and real EHR data in the latent space. The metric first use principle component analysis to reduce the sample dimension for both data and then cluster the samples in the latent space. Ideally, if synthetic and real EHR data are identically distributed, the synthetic and real EHR samples should respectively comprise half of the samples in one cluster. Therefore,

the metric is calculated as:

$$\log\left(\frac{1}{K}\sum_{i=1}^{K}[\frac{n_i^{\mathrm{real}}}{n_i}-0.5]^2\right), \tag{14}$$

where $K$ is the number of resulted clusters, $n_i$ and $n_i^{\mathrm{real}}$ are the sample number and the real sample number in $i$th cluster, respectively. The lower the value, the less synthetic data distribution deviates from the real data distribution. In our experiments, $K$ is decided by the elbow method (Yuan & Yang, 2019) for each synthetic data and in this work is 4 or 5 according to different methods.

### A.2  Medical concept abundance

The metric assesses the synthetic EHR data distribution on the record level. The metric calculates the empirical distribution of the unique positive (occurred) code number within each sample. The empirical distributions are calculated by histograms. The discrepancy between synthetic and real EHR data is calculated as follows:

$$\sum_{i=1}^{M}\frac{1}{2N}|h_r(i)-h_s(i)|, \tag{15}$$

where $M$ is the number of bins in histograms, $N$ denotes the number of samples for real (or synthetic) data, and $h_r(i)$ and $h_s(i)$ respectively represent the $i$th bin in the histograms of real and synthetic EHR data. In this work, M is set to 20.

### A.3  Results

From Table 4, it is shown that EHRDIFF performs better than most baselines and only marginally falls behind medWGAN by 0.922 on the latent distance metric. In terms of MCAD, EHRDIFF consistently outperforms all baselines, and as depicted in Figure 4, we can see that the histogram of unique code count distribution of synthetic EHR data by EHRDIFF achieves the best fit to that of real EHR data. From a sample-level perspective, latent distance results illustrate that synthetic EHR data by EHRDIFF is closely distributed to real EHR data. The MCAD results show that synthetic data by EHRDIFF resembles the real EHR most in terms of unique positive code counts. This result is in line with the findings of the non-zero code column metric.

## B  Data Materials

### B.1  CinC2012 Data

CinC2012 Data (Silva et al., 2012) is a dataset proposed to predict the mortality of ICU patients in the CinC2012. It contains both general descriptors such as age, gender, and ICU type and time series records like heart rate, respiration rate, and serum glucose. In our experiments, we use the preprocessed version of this dataset from (Johnson et al., 2012), which is derived by applying simple extraction on the time-series features and excluding abnormal outliers in the physiological measurements. We then add the label of in-hospital mortality to the records, making 115 features in total. There are 4000 records for model training and another 4000 records for model testing, as split by the CinC authority. We use this dataset to evaluate the models' performance on mixed-type EHR data.

### B.2  PTB-ECG Data

PTB-ECG Data (Bousseljot et al., 1995) is a collection of ECG signals for heart disease diagnosis. We utilize a preprocessed version from (Kachuee et al., 2018) to carry on our experiments, which is segmented and preprocessed from the original PTB Diagnostic ECG Database. The dataset contains 4046 normal patients and 10506 records with heartbeat classified as abnormal. Specifically, all the signals are cropped, downsampled, or padded to make each sample into a fixed dimension of 188. We use this dataset to explore models' ability to generate continuous medical time series data.

