# OpenReview forum: "EHRDiff: Exploring Realistic EHR Synthesis with Diffusion Models"
_TMLR — Rejected by TMLR_

### Review · Reviewer_Wewj · 2023-06-06

**Summary Of Contributions:**

Due to privacy concerns, there is only limited access to large scale and high-quality EHR data. Therefore, it is of importance to synthesize high-quality EHR data. Previous methods focus on Generative Adversarial Networks (GAN) while this paper explores how to use the diffusion model to generate EHR data. This diffusion model-based method outperforms the previous methods and achieves the SoTA performance on this task.

**Audience:**

Yes

**Claims And Evidence:**

Yes

**Requested Changes:**

-	The whole section 3 includes a very general description of the diffusion model, even though this description is necessary but can be shorter as it is not specifically for the EHR.

-	Can the authors discuss how EHR data synthesis is different from vision data synthesis, and how these differences affect the application of the diffusion model on different modalities?

-	The author argues that the EHR data synthesis is beneficial for the case where there is only limited access to the data. However, training the diffusion model itself requires a large amount of data. How does the proposed method compare with the model trained on the diffusion model training data?

-	Following the previous method, how does the performance of the model trained on the synthesized data (e.g. Table 3) change with the amount of synthesized data? Does more synthesized data always result in better results?


**Strengths And Weaknesses:**

Strengths:
- To the best of the reviewer’s knowledge, this is the very first paper to apply diffusion models which often used for visual generation to EHR data.
- The method is effective and generate more diverse data than previously proposed GAN based method.
- The effectiveness of the method is validated on multiple datasets.

Weakness:
- The method is a rather direct application of the current existing diffusion models. The proposed method did not update the original diffusion models designed for vision to the EHR data.

---

> ### Author Response · Authors · 2023-06-14
> **Responses to Reviewer Wewj [1/2]**
>
> Thank you for your valuable comments and useful feedback that have helped us improve our paper.
> We provide responses to the Requested Changes and Weaknesses below:
>
> ```
> 1. The whole section 3 includes a very general description of the diffusion model, even though this description is necessary but can be shorter as it is not specifically for the EHR.
> ```
> Thank you for the advice. We will revise the presentation in Section 3 and may move part of the texts to the Appendix to make the description of the methodology more concise.
>
> ```
> 2. Can the authors discuss how EHR data synthesis is different from vision data synthesis, and how these differences affect the application of the diffusion model on different modalities?
> ```
>  In our opinion, EHR data contains a mixture of data modalities that differ from vision data. In clinical scenarios, EHR data may contain various modalities, including the binary modality such as code features, the categorical modality such as patients' ages, the continuous modality such as test results, or the time-series modality such as electrocardiograms. Although most previous EHR synthesis research focused on binary code data (medGAN Choi et al., 2017; EMR-WGAN Zhang et al., 2020, inter alias), we extend the problem to more real-world scenarios and include different kinds of features in our experiments using various datasets. Additionally, features of different aspects may have different scales, such as ages and heights, while in vision data, channels of different pixels in images share the same ranges. Therefore, different preconditioning methods for the data are required before feeding the whole feature vector into the neural network. Furthermore, in MIMIC III data, binary code features are prone to mode collapse, as the code features have a long-tailed distribution (We will revise the paper and add a demonstrative distribution plot of MIMIC III code data distribution in the Appendix.) and the code feature vector does not have local patterns as image data.  These factors pose challenges and differences to the synthesis of EHR data.
>
> ```
> 3. The author argues that EHR data synthesis is beneficial for the case where there is only limited access to the data. However, training the diffusion model itself requires a large amount of data. How does the proposed method compare with the model trained on the diffusion model training data?
> ```
> The problem with EHR data is that while there is a vast amount of it (maintained by hospitals or healthcare centers), such data is not publicly available due to privacy concerns. This limitation leads to a very small amount of publicly available EHR data for researchers. For EHRDiff or other EHR synthesis models, we consider a practical scenario where hospitals or healthcare centers use their in-house rich EHR data to train an EHR synthesis model and then use the model to generate a large amount of publicly available synthetic EHR data. Some practices of the scenario, such as Synthea (Walonoski et al., 2017), already exist. That motivates us to evaluate EHR synthesis models from both utility and privacy perspectives. Utility focuses on the quality of synthetic EHR data for public downstream research usage, while privacy focuses on the preservation of sensitive information. Good privacy results can ensure the open resource of generated synthetic EHR data.
>
> The downstream models trained with synthetic data from EHRDiff can achieve on-par performance with the models trained on real EHR data which is of the same size as synthetic data and is also the same data used to train EHRDiff, as shown in Table 3. This indicates the great utility of synthetic EHR data generated by EHRDiff.

---

### Review · Reviewer_v2pR · 2023-06-13

**Summary Of Contributions:**

This paper introduces diffusion models directly to unconditional generation of synthetic electronic health records (EHR) data which is publicly scarce due to privacy issues. The authors note that existing methods (especially recent GAN-based ones) for EHR data generation suffer from mode collapse phenomenon and yield unsatisfactory synthetic data quality.

This work is an attempt to bridge this gap by utilizing diffusion models. The authors demonstrate the effectiveness empirically on a real-world public discrete dataset mainly. Generating continuous EHR data is also discussed on two public continuous datasets. Their experiments indicate that diffusion models outperform existing methods in terms of generating synthetic EHR data that is closer to real data distributions. However, it is important to note that diffusion models may also raise higher privacy concerns.

**Audience:**

Yes

**Broader Impact Concerns:**

There is no major concerns on the ethical implications.

**Claims And Evidence:**

Yes

**Requested Changes:**

Please refer to details in above weakness

**Strengths And Weaknesses:**

Strengths:

- The paper is generally well written and easy to follow.
- The empirical results outperform state-of-the-art methods. And the details of proposed method and implementation are solid.

Weaknesses:

- The experiments shows that though diffusion models can generate more realistic data, distinctively higher privacy risks than most existing methods may be caused simultaneously, which is usually unsufferable in real-world applications. As a forerunner of applying diffusion models in HER data generation, it is essential to provide some insights into this gap and the trade-offs between utility and privacy. One possible explanation is that diffusion models are more prone to fit training data distributions. This characteristic may also partly contribute to the superior utility compared with existing methods.
- The paper does not contain many ablative studies. It would be useful to conduct a study on the effect of the design of denoising function and reverse sampling strategy against the utility-privacy trade-off, which corresponds to section 3.4 and 3.3, respectively. These observations may be meaningful materials for interpreting why diffusion models outperform GAN-family methods.
- Given the stochastic nature of diffusion models, it would be appreciated if the paper could report mean and standard deviations of results of multiple runs.

---

> ### Author Response · Authors · 2023-07-04
> **Responses to Reviewer v2pR [1/2]**
>
> Thank you for your precious advises, here are our responses:
>
> ```
> The experiments shows that though diffusion models can generate more realistic data, distinctively higher privacy risks than most existing methods may be caused simultaneously, which is usually unsufferable in real-world applications. As a forerunner of applying diffusion models in EHR data generation, it is essential to provide some insights into this gap and the trade-offs between utility and privacy. One possible explanation is that diffusion models are more prone to fit training data distributions. This characteristic may also partly contribute to the superior utility compared with existing methods.
> ```
>
>
> In light of privacy concerns, our manuscript presents a discussion on the privacy results, as outlined in Section 4.5.2. We agree with the idea that "One possible explanation is that diffusion models are more prone to fit training data distributions. This characteristic may also partly contribute to the superior utility compared with existing methods." This may explain such trade-off, as both privacy metrics utilized in our manuscript consider scenarios where attackers have partial information on an EHR in the training data and seek to obtain the remaining information through synthesized EHR data.
>
> It is evident that if a generative model can produce synthetic EHR data that is perfectly similar to the training data (the synthesized EHR data is the same as training data in an extreme case under utility metrics), attackers can obtain accurate remaining information, leading to the worst results in terms of privacy leakage. Conversely, a generative model with poor utility, producing nonsensical synthesized EHR data, would result excellently in privacy preservation. This is where the implicit trade-off between utility and privacy exists.
>
> On our experiment results, we can see that the privacy results of EHRDiff is worse than medGAN, medWGAN, medBGAN and CorGAN, while these four GAN-based models are also have distinctively worse results in terms of utility as shown in Table 1. When compared to EMRWGAN which achieves the best utility results among GAN-based models, EHRDiff shows less privacy risk in terms of AIR and has on-par performance in terms of MIR. These results empirically demonstrate such trade-off.
>
> ```
> The paper does not contain many ablative studies. It would be useful to conduct a study on the effect of the design of denoising function and reverse sampling strategy against the utility-privacy trade-off, which corresponds to section 3.4 and 3.3, respectively. These observations may be meaningful materials for interpreting why diffusion models outperform GAN-family methods.
> ```
>
> We based our methodology on a general framework proposed by Karras et al. in designing diffusion models. Before obtaining the results in our manuscript, we have experimented with different designs of diffusion settings such as VP and VE as proposed in Karras et al., the different sampling deigns between Euler 1nd method and Heun’s 2nd Method, and we also ablate on different designs of denoising function on whether pre-conditioning is necessary. The results are presented in the following table:
>
> |                   | Corr             | NZC                | CMD             | AIR               | MIR               |
> | -------------- | ------------------ | --------------- | ----------------- | ----------------- | ----------------- |
> | VE             | 0.9876$\pm$0.0003 | 1553.0$\pm$4.967   | 8.057$\pm$0.022 | 0.013$\pm$0.001   | 0.2549$\pm$0.0023 |
> | VP             | 0.9976$\pm$5e-05  | 1564.333$\pm$1.247 | 8.002$\pm$0.013 | 0.0143$\pm$0.0007 | 0.3024$\pm$0.0007 |
> | -------------- | ----------------- | ------------------ | --------------- | ----------------- | ----------------- |
> | Euler’s method | 0.9974$\pm$5e-05  | 1541.333$\pm$4.19  | 7.889$\pm$0.006 | 0.0155$\pm$0.0003 | 0.3111$\pm$0.0012 |
> | -------------- | ----------------- | ------------------ | --------------- | ----------------- | ----------------- |
> | w/o pre-conditioning  | 0.9655$\pm$0.0027 | 432.0$\pm$26.771   | 8.719$\pm$0.058 | 0.003$\pm$0.0009  | 0.3184$\pm$0.0018 |
> | -------------- | ----------------- | ------------------ | --------------- | ----------------- | ----------------- |
> | EHRDiff            | 0.9989$\pm$5e-05  | 1771.0$\pm$1.414   | 7.768$\pm$0.013 | 0.0187$\pm$0.001  | 0.2954$\pm$0.0014 |
>
> Note that Corr, NZC are those metrics where higher values indicates better performance while CMD, AIR and MIR are those metrics where lower values indicates better performance. We can see through the results in the tables that our implemented designs on diffusion processes, sampling methods and denosing functions lead to the best performance of EHRDiff in terms of utility (Corr, NZC, CMD) and maintain good privacy preserving abilities (AIR, MIR).

---

> > ### Author Response · Authors · 2023-07-04
> > **Responses to Reviewer v2pR [2/2]**
> >
> > ```
> > Given the stochastic nature of diffusion models, it would be appreciated if the paper could report mean and standard deviations of results of multiple runs.
> > ```
> >
> > We will report the results of multiple runs in the following revised manuscript, and these parts of experiments are in progress. You can see from the above table where we include standard deviations in the results that the standard deviations are much smaller than the means.

---

### Review · Reviewer_ngsw · 2023-06-20

**Summary Of Contributions:**

In this paper, the authors propose to synthesize EHR data using diffusion models. In particular they make the following contributions:
1. They apply diffusion methods from existing work (Song et al. and Karras et al.) to EHR data.
2. They perform experiments and compare the diffusion approach to GAN- based baselines, using diagnosis and procedure codes in MIMIC-III (a publicly available dataset). The utility and privacy metrics are adopted from previous work. The results show that while the performance of EHRDiff is comparable to existing work in terms of dimension-wise distribution, it has a significantly better result in terms of non-zero code columns, hence it overcomes the issue of mode collapse compared to GAN based methods. However, in terms of attribute inference risk and membership inference risk, GAN based methods perform better.
3. They also test diffusion on two datasets: CinC2012 and PTB-ECG, and show that EHRDiff performs best in terms of AUC.


**Audience:**

Yes

**Broader Impact Concerns:**

There are ethical implications associated with the work so I would recommend adding a Broader Impact statement. Previous work has highlighted issues of generative models and possible biases, which may also occur in EHR data. Hence, a discussion of similar concerns is necessary in the medical context.

**Claims And Evidence:**

No

**Requested Changes:**

Please see the list of weaknesses listed above. They are all critical for securing a positive recommendation for acceptance as the work cannot be accepted in its current form.

**Strengths And Weaknesses:**

**Strengths**
- The paper explores the use of diffusion models for EHR data, which is an interesting and useful application.
- The authors test existing methods on three publicly available datasets: diagnosis and procedure ICD codes in MIMIC-III, CinC2012 Data, and PTB-ECG Data.
- Several metrics are utilized to assess the performance of the models both in terms of utility and privacy.
- Several GAN-based baselines are considered.


**Weaknesses**
- The paper makes a strong claim in that they introduce diffusion models to EHR synthesis. However, based on a quick search there is existing work that also does so, such as: (a) https://arxiv.org/abs/2302.04355 and (b) https://arxiv.org/pdf/2302.14679.pdf The authors also mention this in the introduction "The efficacy of diffusion models on realistic EHR synthesis is rarely studied." They repeat this again in Section 2.2 "Our work for the first time introduces diffusion models to realistic EHR synthesis".
- Privacy as motivation for the work: In the abstract, the authors state that "privacy concerns have resulted in limited access to high-quality and large-scale EHR", however this is not entirely true. Access has improved significantly but data sharing remains difficult. This sentiment is mentioned repeatedly, such as how their proposed work can protect private information. I am not entirely sure that this can be viewed as a privacy-preserving approach, considering that it merely synthesizes medical information, rather than patient identifying information. This is also emphasized in the first paragraph of the introduction. Further discussion is warranted.
- There are several grammar issues throughout the paper, such:
  - introduction paragraph 1: "real-world EHRs need de-identification before publicly accessible"
  -  the use of the term "open-resourcing". Is it a valid term?
  - A line of "works" should be work, similarly for other instances of works
  - "texts"
  - methods "are" proposed
  - "Diffusion models are first proposed and theoretically supported" --> incomplete sentence
  - diffusion models "become" a focused research direction since then
  - There are other issues, such as missing full stops, or "try to infer the rest features".
  - wrong citation: (Agarap, 2018)
- The authors also claim that lack of data sharing impedes precision medicine methodologies - however the latter mostly relies on omics rather than EHR data.
- There is some repetition between the contributions bullet points and the two paragraphs preceding it. The list of contributions also mentions "two following contributions" but they list three
- In the first paragraph of the related work section, the authors mention that existing work is mostly concerned with the generation of ICD codes. Can they provide further details on what the cited studies generate, such as Buczak et al and Walonoski et al, it's not very clear.
- The overall presentation of the work needs improvement, for example Figure 1 is very similar to that in the original work by Song et al, 2021. Can they provide a better depiction of the EHR data rather than the use of very generic icons? Figure captions also need further elaboration.
 - In section 2.2, they use the model names as "DDPM discover the ..." which is not accurate
- In the methodology sections 3.2-3.4, the authors merely redescribed the methodology from existing work, mainly Song et al. 2022  and Karras et al., 2022. Such work must be presented as background work, however the authors refer to it as their own, such as "We brief our 2nd order sampling method", or "we propose a method called EHRDiff" in the contributions section or "to demonstrate the effectiveness of our proposed EHRDiff". The work for the most part applies existing methods, so there are no methodological contributions.
- In the contributions section, the authors claim that "the method can synthesize different forms of EHR data (categorical, continuous and time-series)." However, the main results only show the diagnosis and procedure ICD codes, where each EHR sample is encoded as a fixed size feature vector, where the ith dimension represents the occurrence of the corresponding code feature. Hence, this claim is not supported. They later present some results in the discussion so the current organization of the paper with respect to the proposed contributions is confusing.
- Further information is required on how the data is preprocessed, do they use all ICD codes assigned to a patient stay?
- The data is split into training and and testing, what about validation? There does not seem to be any hyperparameter tuning.
- All baselines are GAN based, it would be interesting to explore other types of diffusion models.
- Figure 2 is referenced before Figure 1. Please reference in order of appearance.
- There are no confidence intervals or statistical testing to clarify significance of improvements. For example, it seems that the performance of EHRDiff is very comparable to EHR-WGAN in terms of dimension-wise distribution (both are 0.999), and in terms of the F1 score in dimension-wise prediction.
- Another claim that is not fully supported: "It is also indicated that synthetic EHR data by EHRDiff may have superior utility for developing downstream biomedical methodologies", with the results presented in the Appendix. Please elaborate / explain what this means as it is a general statement.
- The discussion section presents more results rather than discusses the strengths and limitations of the proposed work. A proper discussion section is needed as it is currently missing.
- the additional experiments on the two datasets are interesting, however there is crucial information that is missing - how were the baselines extended to account for this mixed data, since they were originally developed for coded information. MIMIC also has many benchmarks that can be tested: https://www.nature.com/articles/s41597-019-0103-9

---

> ### Author Response · Authors · 2023-07-04
> **Responses to Reviewer ngsw [1/2]**
>
> Thanks so much for your precious advices, here are our responses:
>
> Regarding the claims made in our paper, we acknowledge that our work was appeared after the studies mentioned in your comment. Although we did not submit our manuscript to ArXiv before these studies, we finished our study before both works and we did cite MedDiff in our paper. We agree that the claim of being the "first research" in this area should be revised. However, we believe that our contribution of applying diffusion models to EHR synthesis is still significant, given the limited research on this topic compared to GAN-based models. Our results demonstrate the potential of diffusion models in EHR synthesis, without the need for additional complex sampling procedures proposed in MedDiff. Therefore, we believe our work provides a strong baseline for future research in this area.
>
> On the topic of privacy concerns, we agree that access to real EHR data has improved with the release of MIMIC IV. However, the publicly available EHR data is still limited and only includes a few types of EHR data, such as mostly discharge summaries for ICU in MIMIC. Therefore, we believe that EHR synthesizing methods are still useful. As for the privacy evaluation, we used de-identified EHR data for our experiments. While our model synthesizes medical information, we acknowledge that there may be potential privacy risks. To evaluate the privacy preserving ability of our generative models, we followed the evaluation procedure proposed in a previous study titled "A multifaceted benchmarking of synthetic electronic health record generation models," published in Nature Communications. We detailed our privacy evaluation methodology in Section 4.3.2, which considers the privacy attacking risks from different perspectives.
>
> Regarding grammatical issues and inaccurate presentation, we will revise the typos and grammatical errors in our manuscripts. Additionally, we will review our presentation to ensure accuracy, including:
> - Changing the expression “precision medicine methodologies” to “phenotyping, automatic diagnosis etc” which requires EHR data for methodology development.
> - Clarifying that Buczak et al designed it to generate ICD codes, while Walonoski et al proposed Synthea, which built an EHR synthesis pipeline using a combination of generative models to generate mixed types of EHR features.
> - Providing the full name of DDPM in Section 2.2;
> - Detailing the data preprocessing procedure. In Section 4.1, we provide a detailed description of our data preprocessing procedure. This includes using all the ICD codes of diseases and procedures assigned to a patient's admission, as well as truncating the code digits to reduce the long-tailed distribution of codes and the size of the code set.
> - Revise the order of figure reference to match their appearance order, and adding more detailed explanation to Figure 1 and its caption.
>
> We will also revise our presentation of both our contribution claims and methodology. While it is true that "The work for the most part applies existing methods”, implementing diffusion models directly into EHR synthesis is not a straightforward task, as demonstrated in the experiments of MedDiff where their vanilla implementation of DDPM harvested bad generation results on EHR synthesis. Our proposed EHRDiff, which employs different neural networks for denoising functions and varying parameters designs for the diffusion process, outperforms GAN-based methods in generating EHR data for MIMIC codes. In the distribution section, we would move the results on Cinc2012 and ECG data to the main results section which will back up our claims “the method can synthesize different forms of EHR data”.  For a discussion part, we are currently conducting experiments on ablating the denoising function designs and the diffusion processes and on the scaling performance of synthetic EHR data as commented by Reviewer Wewj, this would serve as a proper discussion results and enrich the contents and insights provided in our paper.

---

> > ### Author Response · Authors · 2023-07-04
> > **Responses to Reviewer ngsw [2/2]**
> >
> > In EHR data synthesis, the dataset is split into training and testing sets. The models are trained using the training split and the resulting trained model is used to generate synthetic data. The quality of the synthetic data is then evaluated against the training data split as the generative model is designed to fit the distribution of unsupervised training data, and we search for some hyper-parameter designs based on the metrics such as dimension-wise distribution. Unlike vision data, assessing the quality of synthetic EHR data on the sample level is not straightforward. Therefore, we follow previous research and evaluate the synthetic data against the training data to determine how well the synthetic data fits the data distribution. For downstream models used in dimension-wise prediction and CinC2012 and ECG data, we use the same training setting for synthetic EHR data from different generative models to ensure a fair comparison. We do not conduct hyper-parameter searching for downstream model training.
> >
> > Regarding the baseline selection, all baselines are currently GAN-based. However, we admit “it would be interesting to explore other types of diffusion models”. We argue that the direct implementation of diffusion models is non-trivial and existing state-of-the-art EHR synthesis models are GAN-based. Although MedDiff is a concurrent diffusion model, they have not shared their code and experiment settings for re-implementation.
> >
> > Regarding the confidence interval of our results, we will add the standard deviation for our main results to our manuscript during revision. Under the dimension-wise distribution, observed improvement is not significant; however, the correlation may be biased by codes with a high prevalence, while codes with very low prevalence have limited leverages on the correlation score. This is demonstrated by the non-zero column metric, where EMRWGAN fails to generate many low-prevalence codes, yet still achieves high correlation scores. In the context of dimension-wise prediction, although EMRWGAN demonstrated comparable performance in terms of correlation, we observe that the points from EHRDiff are closer to the diagonal lines in the figure.
> >
> > Regarding the statement "It is also indicated that synthetic EHR data by EHRDiff may have superior utility for developing downstream biomedical methodologies," we suggest that the synthetic data from EHRDiff can be advantageous for training downstream biomedical models, enabling them to achieve comparable performance to models trained with real data. This reflects the good utility of EHRDiff synthetic EHR data. We intend to revise this statement to be more precise.
> >
> > Regarding the ethical implications of our research, all EHR data used in our study are publicly available and de-identified. Our research provides a methodology for synthesizing EHR data, there could be potential of abusing of the model like other generative modeling methods and privacy leakage. Although the model performed exceptionally well on our benchmarks, there is still room for improvement in synthesizing EHR data with more comprehensive information.

---

### Decision · Action_Editors · 2023-08-07

**Recommendation:** Reject

**Comment:**

Some of the concerns raised by the reviewers require addressing directly in the paper, and are quite major. Please revise the paper as the reviewers suggested, in particular:
- Please update the discussion to more accurately position your paper.
- Please improve the presentation of the background work.
- Please include confidence intervals when appropriate.
- Please revise the "discussion" section as indicated by the reviewer ngsw.
- Please include the additional results requested by the reviewer v2pR.
- Please include the discussion points brought up by the reviewer Wewj.

**Audience:**

This paper is a bit of an outlier for TMLR, however, it would be of interest to some. I am always a little conflicted about papers like that, however, after a revision, given that we review primarily for correctness, I anticipate it should be accepted.

**Claims And Evidence:**

This is not a bad paper, as far as I can tell, most of its claims are supported by evidence and correct, however as the reviewers pointed out, most convincingly the reviewer ngsw, the paper needs a bit of restructuring. I think it should be rejected, and then likely accepted after a revision.

**Resubmission Of Major Revision:**

The authors may consider submitting a major revision at a later time.